# Material Property Recovery by Controlling the Melt Memory Effects on Recrystallization and on Crystal Deformation: An Approach by the Molecular Dynamics Simulation for Polyethylene

**DOI:** 10.3390/polym14153082

**Published:** 2022-07-29

**Authors:** Takashi Yamamoto, Mohammed Althaf Hussain, Shigeru Yao

**Affiliations:** 1Graduate School of Science and Engineering, Yamaguchi University, Yamaguchi 753-8512, Japan; 2Faculty of Technology, Fukuoka University, Fukuoka 814-0180, Japan; altaf.mh7@gmail.com (M.A.H.); shyao@fukuoka-u.ac.jp (S.Y.)

**Keywords:** plastic recycle, melt memory effects, recrystallization, large deformation, molecular dynamics simulation

## Abstract

Degradation in the mechanical properties of recycled polymer materials has been recently appearing as a big issue in polymer science. The molecular mechanism of the degradation is considered in part due to residual memories of flow experienced during molding processes, and therefore the mechanical recycling through remolding involving melting and recrystallization has been attempted in recent years. In the present paper, the molecular processes of melting and recrystallization are investigated by the molecular dynamics simulation for polyethylene with special interest in the melt memory effects. We also studied the mechanical properties of the recrystalized materials that have undergone different recrystallization processes aiming to discover better recycling strategies. A successive step-by-step approach is adopted to study the loss of the crystal memory during retention in the melt, the effects of the melt memory on the mode of recrystallization, the relation between the recrystallization mode and the resulting higher-order structure, and the mechanical properties controlled by the higher-order structures. It is shown that the melt memory clearly remains in various order parameters that persist over time scales corresponding to the Rouse time, the remaining melt memory markedly affects the crystallization mode leading to distinct crystalline morphologies, and the distinct morphologies obtained give different mechanical responses during large deformations.

## 1. Introduction

Lower mechanical properties in recycled polymer materials are due to various external conditions, such as temperature, stress, ultraviolet rays, etc., during their usage before recycling, where degradation in molecular and/or higher order level, as well as the residual foreign substances, are considered the origins. Understanding the mechanism of the deterioration in physical properties due to these factors has recently become a large issue in polymer science [1].

Elucidating a wide variety of physical degradation processes at the molecular level and clarifying their basic principles are research subjects that form the basis of developing new methods of controlling the physical property of recycled materials. Mechanical recycling is accomplished through a remolding process that involves the melting of the used plastics and recrystallization into recycled ones [2].

Regarding the melting of the plastics, the history of the initial molding process is considered to remain even after melting, and this residual history of the so-called melt memory has a significant influence on the recrystallization [3]. However detailed molecular process of such recrystallization from the melt with residual memory is still a large challenge [4].

Many experimental studies have long been reported regarding the melt memory effects on crystallization [5]. Some “structural memory” remaining in the melt is considered to have a great impact on crystallization; even in the seemingly uniform melt state, we are forced to consider that some spatial structures remain. Indeed, recent experimental studies using thermal analysis, for example, indicated great acceleration in crystal nucleation suggesting the presence of some embryonic structure in the melt [5].

However physical entity of the memories are considered local and transient, and there are great difficulties in directly identifying their molecular images due to the limited spatial and temporal resolutions. On the other hand, some previous studies suggested the importance of molecular entanglements [6] instead of the local order; the entanglements in the original crystalline texture remain even after melting and they are considered to give great influence on recrystallization. However, here again, experimental evidence is difficult to obtain; the entanglements are not the thermodynamic substances but topological ones with major effects on chain kinetics, and the molecular-level reality is generally not well resolved.

What is the molecular mechanism of retaining the melt memory of the previous history? How does it affect the material’s recrystallization during the molding process? The molecular mechanism of the melt retaining the memory of previous history, and the way it greatly affects the material’s recrystallization during molding process are our major subjects of interest. Our present goal is to have a unified understanding of recrystallization and higher-order structure formation and to elucidate the origin of the reduced mechanical properties in recycled materials. As described before, various factors can cause degradation of the plastic materials also including environmental influences causing chemical degradation during their usage.

However, our present work only focuses on the physical processes of higher order structure formation during recrystallization and the resulting alteration of the mechanical properties. For this purpose, our powerful research tool is computer modeling, which has been rapidly developed in recent years [7]. It will enable direct observation of polymer recrystallization and deformation at the molecular level [8,9,10]. We would like to elucidate the fundamental principles behind the complex phenomena aiming for the innovation in the materials recycling. We will hereafter make a step-by-step approach focusing on the following points: (1) loss of the crystal memory during retention in the melt, (2) relationship between the melt memory and the mode of recrystallization [5], (3) relation between the mode of recrystallization and the higher-order structure formation, and (4) the higher-order structure’s control of the mechanical properties during large deformation and fracture [11].

We here investigate polyethylene, the simplest polymer, as our starting material for the molecular dynamics (MD) investigations of the mechanical recycling of crystalline polymers. Although many experimental investigations have already reported on the PE recycling, there is no systematic approach, as far as the authors are aware, based on the molecular level simulations of recrystallization and deformation.

## 2. Models and Methods

### 2.1. Modeling and Simulations

In this paper, we study PE, as in our previous papers [8,12], using a conventional united atom model combining every hydrogen atom to its nearest carbon atom. We consider 150 or 300 PE molecules each comprising 500 methylene groups; the total number of the united atoms in the system is either 75,000 or 150,000. Taking the consistency of the present research with our previous works, we here adopt the Rigby–Roe force field [13], which consists of the following intramolecular energy terms: C-C bond-stretching ($U_{bond}$), C-C-C bond-angle-bending ($U_{angle}$), and dihedral-angle rotation ($U_{torsion}$),
(1)$$U_{bond}\left(r\right)=k_{b}{\left(r-r_{0}\right)}^{2}/2$$
(2)$$\;\;U_{angle}\left(\theta \right)=k_{\theta }{\left(cos\theta -cos\theta_{0}\right)}^{2}/2$$
(3)$$U_{torsion}\left(\tau \right)=k\overset{5}{\underset{n=0}{\sum }}a_{n}cos^{n}\tau$$
where $r_{0}$ is the equilibrium bond length 0.152 nm, $\theta_{0}$ is the equilibrium bond-angle of 70.5 deg., and the torsion angle τ is measured from the trans position. The nonbonded interactions $U_{vdW}$ of the following form are assumed between united atoms of different chains and between those of the same chain more than three bonds apart,
(4)$$U_{\mathrm{vdW}}\left(r\right)=4\varepsilon \left\{{\left(\frac{\sigma }{r}\right)}^{12}-{\left(\frac{\sigma }{r}\right)}^{6}\right\}$$
where ε and σ are assumed to have values 0.5 kJ/mol and 0.38 nm, respectively; the interactions are cutoff at a distance $r_{c}$ = 2.5σ, and no tail corrections are made. The numerical values of the parameters and the related reduced units are reproduced in Table 1 for convenience.

We placed PE molecules within a rectangular parallelepiped MD cell having three sides of lengths (*a*, *b*, and *c*), which is replicated by the periodic boundary condition (PBC). Most of the MD simulations are done using the program LAMMPS (large-scale atomic/molecular massively parallel simulator) [14] with the aid of COGNAC (coarse-grained molecular dynamics program) in OCTA (the open computational tool for advanced materials technology) [15] as a pre-post processor for the LAMMPS. The temperature and pressure are controlled mostly by the conventional NTP ensemble in the LAMMPS.

### 2.2. Analysis of Melt Order, Crystallinity, and Chain Entanglements

In order to capture the emerging crystalline order, we must introduce several order parameters. We first define so-called chord vectors $b_{i}$, which are defined as connecting the mid points of the two adjacent C-C bonds, corresponding to the i-th united atoms located at $r_{i}$**.** We can here define a local order parameter $P_{2}\left(r\right)$, which has been used in our previous papers [12], by dividing the MD cell into cubic mesh-cells of side length 2*σ* (volume 8$\sigma^{3}$), and within each mesh-cell, we calculate the orientational order parameter,
(5)$$P_{2}\left(r\right)=\langle 3\cos^{2}\theta_{i,j}-1\rangle /2$$
where $\theta_{ij}$ is the angle between chord vectors $b_{i}$ and $b_{j}$ within the same mesh-cell located at the representative position ***r***, and the average is taken over all pairs of chord vectors within the mesh-cell. Then, we estimate the crystallinity $x_{c}^{mesh}$ from the number of mesh-cells $N_{c}$ that have local order parameter greater than a threshold 0.7 divided by a total number of mesh-cells in the system $N_{total}$
(6)$$\;x_{c}^{mesh}=N_{c}(P_{2}>0.7)/N_{total}$$

The choice of the threshold 0.7, which we used in our previous simulations, might seem rather small; however, a slightly stringent threshold did not greatly affect the results. We also use the orientational order parameter, for example $P_{2z}$ of the chord vectors against the *Z*-axis direction,
(7)$$P_{2z}=\langle 3\cos^{2}\theta_{zi}-1\rangle /2$$
where $\theta_{zi}$ represents the angle between the chord vector and the *Z*-axis, and the average is either over all chord vectors in the whole system or over the prescribed space, such as crystalline regions. Similar order parameters can be defined using the end-to-end vectors of the chains.

During the rapid formation of crystallites, we expect the pronounced generation of folds and ties connecting the crystallites together with dangling tails (cilia). We here regard the crystallizing system as a network composed of the crystallites, ties, folds, and cilia. In order to describe the network structure, we define basic network components as given in our previous work [12]. Since this subject is based on the molecular level details of the crystals, we must redefine the crystal clusters in some detail. The criterion that we used as for the crystalline atoms is similar to what we have used before [12]; we first pick up straight chain segments that are made of at least nine chord vectors whose angle between neighboring chord-vectors are less than 30 deg., and we consider these segments as crystal candidates; of these candidates, we consider those having more than two parallel and neighboring stems to be crystalline.

This crystal criterion may be slightly weaker and resulting crystallinity $x_{c}^{atomic}$ is a slightly larger than that we defined before using crystalline mesh-cells; however, the discussions are not affected by the change of the definition. Having defined the crystalline atoms in this way, we can categorize the connecting segments as ties, folds, and cilia according to the rule shown in our previous work [12].

We also studied the possible effects of chain entanglement in the melt on recrystallization; we also calculated the entanglements remaining in the solid state, such as the so-called trapped entanglements between folds. Several methods have been proposed for counting the number of entanglements, and we here use the Z1 code proposed and made available by Kröger [16].

## 3. Results of Simulations

### 3.1. Disappearance of the Crystal Memory during Retention in the Melt

First, we aimed to clarify the origin of the melt memory by directly observing the structure of the melt, which was generated by constant heating of the stacked lamellae. Figure 1a shows our starting crystalline structure, which was obtained in our previous MD simulation of fiber formation from the stretched PE melt [12]. In the figure, the white areas are crystalline regions where the polymer chains are oriented in the *Z*-axis direction of stretching (horizontal), and the inter-crystalline regions are filled with folds or loops (yellow), ties or bridges (red), and cilia or tails (green).

This starting crystalline system was melted by a constant heating (10 K/ns), and the melting process was investigated. In order to quantitatively evaluate the melting process, we used the crystallinity $x_{c}^{mesh}$. Figure 1b shows the temperature changes in $x_{c}^{mesh}$ during heating both under stress free condition ($P_{xx},\;P_{yy},\;P_{zz}$) = (0, 0, 0) (black) and that under constant tension condition ($P_{xx},\;P_{yy},\;P_{zz}$) = (0, 0, 1) (red); in both cases, the crystals were found to be completely melted around 440 K irrespective of the stress condition; by much slower heating; however, the equilibrium melting temperature ($T_{m})$ was estimated to be between 420 and 430 K.

In Figure 1d, we show the decrease in the order pa rameters $P_{2z}$ for the chord vectors {$b_{j}$} during the constant heating; the order parameters $P_{2z}$ show rapid decreases around the melting temperature, similar to the crystallinity; however, they remain finite even around 440 K where the crystallinity disappears completely. Figure 1c is the molecular graphics of the crystallites during melting under tension free condition; the crystalline regions are here painted in blue and the amorphous regions in red, where the melting begins around 400 K with the lamella crystals gradually becoming thinner by surface melting and the crystal edges (the side surfaces of the lamellae) receding.

An essential question in the melt memory effect is how long the melt memory remains. The time period that the melt memory remains will largely depend on the holding temperature $T_{h}$ (strictly speaking, the superheating ∆T = $T_{h}$ − $T_{m}$> 0); however, here, we only consider the case $T_{h}$ = 440 K (∆T~10 K). Figure 2a shows the changes with time in the snapshots of the melt when held at 440 K under stress free condition. The small blue dots represent are the residual small crystalline areas or embryos, which gradually decrease with time. The system shows large shortening in the *Z*-axis direction during retention in the melt suggesting that the initial stretched molecules in the melt are gradually relaxing and shortening in the *Z*-axis direction.

Figure 2b is the orientational order $P_{2z}$, both of the chord vectors (black dots) and of the end-to-end vectors (red dots), which show the marked loss of the melt memory during the time $t_{hold}$ held at 440 K. Both order parameters significantly reduced within a few nanoseconds, which may be related to the Rouse time $\tau_{R}$ for the highly stretched polyethylene of length 500 carbons; of special interest is that the decrease of $P_{2z}$ defined from the chord vectors shows clear exponential decrease as indicated in the red circles in Figure 2c. We also studied the changes in the radius of gyration $R_{g}$ of the chains (Figure 2d); the molecules initially highly stretched along the *Z*-axis rapidly shrink and relax to an almost equilibrated random coil state in around 10 ns.

In addition to these $P_{2z}\;$ and $R_{g}$, we also studied the behavior of other order parameters, such as the trans sequence lengths $l_{t}$ and their distribution $P\left(l_{t}\right)$, the averaged persistence length $\left\langle L_{p}\right\rangle$, as well as the degree of shrinkage of the radius of gyration ($R_{gz}/R_{gz0})$ where $R_{gz0}$ is the radius only at the start of holding at 440 K. All these parameters for the melt state showed clear relaxation toward the equilibrium within a few nanoseconds, and they appear to have close correlations with $P_{2z}$.

The local orientation order $P_{2z}$ of the chord vectors was shown to be a main origin of the accelerated crystal nucleation under flow [17] as will be also described in the following sections. However, the molecular mechanism of such local order in the melt dominating the nucleation or/and interfacial processes at the nucleus melt interfaces remains a mystery; the real molecular mechanism of the rapid nucleation is not easy to confirm.

Extensive research has long been done on the polymer entanglements in the melt especially focusing on rheological problems, where the number of entanglements Z per chain and the segment lengths between the adjacent entanglements are discussed. We investigated the number of entanglements Z in the melt and their possible roles in the process crystallization by using the Z1-code proposed by Kröger [16]. Figure 3 shows the relation between the number of entanglements per chain Z and the retention time $t_{hold}$ in the melt held at 440 K.

The tendency for the number Z to slightly increase with $t_{hold}$ is noticeable, which could be due to the re-entanglement of the partially disentangled melt in the initial drawn melt. However, the increase in the entanglement number Z appears to be too small to explain the remarkable changes in the crystallization rate with $t_{hold},$ which will be described later; the effect of chain entanglements on crystallization does not appear to be important in the melt memory effects that we are considering here.

### 3.2. Crystalline Memory in the Melt and the Recrystallization Rate

The ample memory is confirmed to remain in the melt. Then, the way it affects recrystallization is the next important question. Figure 4a shows the time dependences of the degree of crystallinity $x_{c}^{mesh}$ during the early nucleation stages; the system was rapidly cooled to the crystallization temperature (350 K) from the melt, which was held for various times $t_{hold}$ at 440 K. The early stage crystallization, or the nucleation rate, is shown to depend sensitively on the retention time $t_{hold}\;$ at 440 K.

Figure 4b shows the evaluated initial slopes of Figure 4a vs. the local bond orientation order $P_{2z}$ given in Figure 2b, where we can find a clear correlation between the logarithm of the obtained nucleation rate and the bond orientation order $P_{2z}$ in the melt. This result nicely corresponds to a recent MD study showing the definite correlation between the nucleation rate *I* and the degree of chain elongation in the melt $P_{2z}$, where $log\left(I\right)$ was shown to be proportional to the local orientation order parameter $\;P_{2z}$ [17].

We then considered the later stages of the crystal growth. Figure 5a shows the crystallinity vs. crystallization time over several hundred of nanoseconds. The figure clearly shows that the crystallization is rapid when sufficient melt memory is retained in the case $t_{hold}=1\sim 3\;\mathrm{ns}$, while it becomes significantly slower for $t_{hold}=4\;\mathrm{ns}$ and actually becomes invisible when $t_{hold}$ is 5 ns. The changes in the melt structure causing such a large reduction in crystallization rate is very interesting. Figure 5b is a log–log plot of the same crystallinity data, a so-called Avrami plot.

The crystallinity $x_{c}$ is generally expressed as a function of crystallization time *t* as $x_{c}=1-\exp\left(Kt^{n}\right)$, and it reduces to a simple power law $x_{c}\sim Kt^{n}$ unless the growing crystals severely overlap; the log–log plot then becomes a straight line, and its slope *n,* called the Avrami exponent, represents the spatial form of the growing crystals and the birth rate of the nucleus. Figure 5b clearly shows that the exponent *n* is about 1 when the memory remains strong ($t_{hold}=1\;\mathrm{ns})$; however, the value of *n* increases from 2 to 3 as the memory is being lost ($t_{hold}$ = 2~4 ns). This indicates that, as the melt memory is being lost, the mode of crystal growth relaxes to the normal three-dimensional case; here, we assumed that there are nearly constant number of embryos and neglect the time dependence of their numbers.

### 3.3. The Melt Memory and the Morphology Development in Recrystallization 

The variation of the Avrami exponent *n* with increasing retention time $t_{hold}$ in Figure 5b suggests the change in the shape of the growing crystals by the relaxation of the melt structure with time $t_{hold}$ held in the melt. We show, in Figure 6, the shapes of growing crystals using the molecular graphics images. Through the growth modes, the dimensionality of the crystals, are not readily noticed from the figures, the fast growing crystals for $t_{hold}=1\;\mathrm{ns}$ with *n* = 1 (Figure 6a) can be seen growing predominantly in one dimension along the crystal thickness direction, while those for $t_{hold}$ = 2 and 3 ns with the exponent *n*~2 (Figure 6b,c) indicate the dominance of the lateral growth along the X–Y directions representing the usual mode of growth in the chain-folded lamellae.

The last case of the slowest possible growth $t_{hold}$ = 4 with the index n~3 (Figure 6d) is predominantly 3D growth of the crystal; such a peculiar 3D growth mode for the chain folded crystal was already noticed in our previous MD simulation [12] where the usual lateral growth proceeds simultaneously with crystal thickening as least for the early stage growth before inter-lamellar collisions. In addition to the changes in the 3D shape of the growing crystals, we also noticed considerable tilt of the crystalline chains.

The snapshots given in the lowest row of Figure 6 show the final crystalline texture obtained after sufficient time of crystallization, where we notice prominent morphological changes in the texture and the orientations of the crystalline chains; the crystalline chains highly oriented along the *Z*-axis (the direction of the initial elongation) gradually lose their original preference and begin to show considerable chain tilting from the *Z*-axis.

Such large changes in the crystalline texture are expected to give distinct mechanical response to the elongation. Figure 7 shows the *Z*-axis elongation for the two representative textures (upper) and their mechanical properties (lower); the two textures (a) and (b) correspond to the state A and the state B defined in Figure 5a for the retention times of $t_{hold}$ = 1 and 4 ns. The degree of uniaxial deformation along the *Z*-axis (horizontal) is represented as the ratio of the length $L_{z}$ against its original length $L_{z0}$. In the state A (Figure 7a), the crystals form the stacked lamellae and the molecules in the crystals retain dominant *Z*-axis orientation reflecting the strong Z-orientation remaining in the melt.

On the other hand, in state B (Figure 7c), the 3D shapes of the crystals are nearly elongated ellipsoids, and the orientations of the crystalline chains show great deviation from the *Z*-axis, which corresponds to the considerable loss of the melt memory. Our preliminary studies of the stress–strain relations for the well-developed crystalline textures shows that the highly oriented crystalline structure recrystallized under strong memory gives clear cavities in the amorphous regions (Figure 7a), which is likely due to the strong lateral constraint, leading to the brittle fracture and rapid stress drop (Figure 7b).

On the other hand, the well-crystallized sample from the largely relaxed melt shows the crystalline texture with largely tilted chains, whose *Z*-axis elongation induced considerable crystal slip and molecular reorientation leading to the nearly constant volume deformation (Figure 7c) and the stress–strain curve showing plastic flow (Figure 7d). The relaxation of the flow memory and of the chain orientations are suggested to give reduced crystal orientations along the *Z*-axis, resulting in the facile plastic deformation; this can be a mechanism by which the sample comes to have high toughness, the details of which processes will be discussed in the following section.

### 3.4. Lamella Morphology and the Mechanism of Large Deformation

In the previous discussions, the varied degree of melt memory with time $t_{hold}$ in the melt resulted in remarkable alteration in the mode of recrystallization, and in the crystal morphology and mechanical property. The molecular mechanism of deformation is expected to offer reasonable explanations for the improved mechanical properties of the materials crystallized from the relaxed melt. However, the relationship between the crystalline texture and their mechanical properties, such as the stress–strain relationship or fracture under large deformation, is a very difficult task to clarify due to the large complexity of the lamellar structure of crystalline polymers.

Figure 8a is a simple picture of a spherulite and the possible modes of deformation. Crystalline lamellae in the spherulite are extending radially from the spherulite center and aggregating to fill the whole spherical region. Uniaxial elongation of the spherulite along a direction (vertical in Figure 8a) gives distinct deformation for the lamellae depending on their orientation against the draw direction. Very long polymer chains connecting the crystal regions give many complicating factors for the deformation.

We here consider the two limiting types of the deformation of lamellar crystals, one along the crystal chain axis (type P in Figure 8c) Pand the other transverse to the chain axis (type T in Figure 8b). Since the crystalline chains are arranged nearly perpendicular to the lamella plane, the mechanical properties of the crystals are extremely anisotropic, and therefore the polymer crystal deforms in completely different ways for these two types of deformation. We consider sufficiently large crystals made of 150,000 united atoms, where the fully crystalline states are generated in the same way as we used in our previous work [12].

In the following subsections, we investigate the large deformation of the two type, where the deformation was constant rate elongation of the true strain rate (the Hencky-strain rate) of 0.05/ns; $\varepsilon_{H}=0.05\times t\left[ns\right]$.

#### 3.4.1. Uniaxial Elongation Parallel to the Chain-Axis (TYPE P Elongation)

Let us first consider the uniaxial deformation along the crystal chain axis as shown in Figure 8c. The stacked lamellar crystals are dawn parallel to the chain axis perpendicular to the lamellar plane (Figure 9a), where the relatively small macroscopic deformation is found to give rise to the formation of cavities in the amorphous regions. With progressive deformation, the cavity grows in size and extends in the chain axis (*Z*-axis) direction. We also notice the formation of fibrils extending along the *Z*-axis on both sides of the cavity.

Such a structure, consisting of cavities and neighboring fibrils, resembles that of the craze observed in the brittle fracture of amorphous polymers. When a cavity starts growing from one place, the macroscopic deformation is mainly due to the cavity with the deformation in other amorphous regions being small. The stress–strain curve in this case gives only small elongation before a sudden brittle fracture by the elongation of about 5% (Figure 9b). One of our concern in this mode of deformation is on the structural changes in the amorphous phase. Figure 9c,d show the molecular level details in the vicinity of the cavity, particularly focusing on the structures of folds and ties.

Within an amorphous layer, the folds that are rooted on the opposite lamellar surfaces generally do not form the so-called trapped entanglements; this is readily noticed from Figure 9c where many folds around the cavity are seen to be well-separated.

This finding may appear somewhat strange and against previous simulation reports [18]. However, considering the initial highly stretched melt conformations and the following crystallization by reeling in the chain tails to the crystalline regions [12], there may not ample opportunity to form trapped entanglements between the neighboring lamellae; in fact, the entanglement analysis between the folds using the Z1 analysis gave only 10% of entanglements per fold (one entanglement per ten folds). In a similar way, Figure 9d shows that the tie molecules seem to be pushed aside by the cavity and many of the tie molecules finally form fibrils connecting the adjacent lamellae; the abundant fibrils can serve as the residual stress transmitting components after the brittle fracture.

#### 3.4.2. Uniaxial Elongation Transverse to the Chain Axis (Type T Elongation)

In the transverse drawing perpendicular to the chain axis (Figure 8b), the crystalline chains initially oriented along the *Z*-axis accomplish large-scale reorientation of about 90° toward the *X*-axis direction [8]. Figure 10 shows the typical molecular processes of the large deformation and reorientation.

In the early stage of deformation, we first notice the large reorientation of the crystalline chains toward the *X*-axis, which inevitably leads to the thinning of the lamellae along the *Z*-axis. The crystallites with largely tilted chains give rise to facile shear deformation by significant crystalline chain slips; the macroscopic deformation is nearly constant volume process without cavitation. The chain reorientation and crystal reorganization, which is often called “crystal breaking” or “mechanical melting”, is particularly conspicuous until 30 ns, after which many fibrils grow along the *X*-axis.

We here investigate the molecular processes of the transverse elongation by using various physical parameters: the level of stress, the volume of the MD cell, the crystallinity, and the degree of chain reorientations both in the crystalline and in the amorphous regions (Figure 11). In the very early stage up to 10 ns where the Hencky strain is less than 0.5, the system shows soft elasticity and yield process.

This is then followed by an elongation under nearly constant stress level up to around 30 ns ($\varepsilon_{H}$~1.5) (Figure 11a), during which, the volume of the system shows slight increase suggesting the birth of small cavity-like regions (Figure 11b); also noticed is the decrease in the crystallinity usually referred to as the “crystal breaking” or the “mechanical melting”(Figure 11c). This stage is considered as the crystal deformation with simultaneous crystal breaking and chain reorientation, where the original stacked lamella structure is greatly perturbed.

The degree of chain reorientation during this stage is clearly seen in Figure 11d, where it is noticed that the chain orientation along the *X*-axis and that along the *Z*-axis nearly replace giving complete (90 deg.) conversion of the crystal orientations.

The crystalline regions show more marked reorientation than the amorphous regions; the amorphous chains may be forced to reorient driven by the large crystal reorientation by the chain slips. After these large-scale reorientation and crystal reorganization, there follows the strain-hardening process between 30 ns < t < 60 ns (1.5 <$\;\varepsilon_{H}$ <3) as observed in Figure 11a, where the crystallinity nearly recovers (Figure 11c). The chain orientations gradually improve further (Figure 11d) until the whole system attains the final fiber structure fully reoriented to the *X*-axis direction (Figure 11d).

## 4. Discussion and Conclusions

The oriented melt of the present polyethylene of 500 united atoms, though very short, was found to keep appreciable melt memory in the chain orientation at 440 K for about 5 ns. The local melt memory was found to show large variations with the retention time, by use of various variables, such as the bond orientation order $P_{2z}$, the radius of gyration $R_{g}$, the persistence length $L_{p}$, and the trans sequence length distribution, etc. Although the relaxation time of about 5 ns may seem rather short, recent MD simulations for PE of the similar chain length showed the Rouse time of the same order of magnitude [19]. On the other hand, the number of chain entanglements showed only a slight increase with the retention time $t_{hold}$, where the re-entanglement was considered negligible during the process of losing memory of the initial stretching in the melt.

These structural memory remaining in the melt showed remarkable acceleration in the nucleation rate and the changes in the crystal growth mode, which is consistent with experimental observation of large changes in the crystallization rate with superheating ∆T = $T_{h}$ − $T_{m}\;$> 0 and/or holding temperatures [5]. Furthermore, the local orientation order parameter $P_{2z}$ in the melt was suggested to have a clear correlation with the primary nucleation rate, which was also indicated by a previous MD work [17].

At the same time, the growth rate of the crystals after the primary nucleation was found to depend considerably on the degree of melt memory (the retention time at the melt state); the mode of crystal growth expressed in terms of the Virial exponent was shown to vary from about 1.0 to 4.0 during the process of losing melt memory; with the loss of memory the mode of crystal growth changes from the one-dimensional along the draw direction to the three dimensional one.

At the same time, the crystalline texture in terms of the chain orientation in the crystalline regions responded systematically to the melt memory, where the chain orientation in the crystalline regions gradually lost their initial *Z*-axis orientation, thus, giving more isotropic chain orientation in the crystals. These findings were also consistent with our previous MD simulation of crystallization from the melts under various tensions along the Z-direction [12]. Higher chain orientation remaining in the melt gave rise to easier primary nucleation and a larger number of remaining crystal embryos, and also to the growth of crystals with a higher chain orientation. The effects of chain entanglement that may be expected as an origin of the melt memory were, at least in our present model of short chain length, found to be small—not giving sufficient contributions to the crystallization.

We also found that the different crystalline textures obtained gave quite distinct mechanical responses to the elongation. The highly ordered stacked lamellae grown from the melt with higher melt memory resulted in the larger initial modulus but followed by the brittle facture with pronounced emergence of voids (cavities) in the amorphous regions. While the crystals of less aligned chains showed much larger plastic deformation under nearly constant volume condition at much lower stress levels, which was considered due to much easier chain slip in the crystalline regions.

The crystalline texture and the way the crystalline chains align in each crystal domain were considered to dominate the mode of macroscopic deformation. We investigated the two extreme cases of the crystalline textures. We found that the elongation parallel to the crystalline chains (transverse to the long axis of the lamellae) was the brittle fracture with pronounced void formation, while the transverse elongation (parallel to the lamellar direction) was the remarkable plastic deformation and large scale chain-reorientation resulting in the well-developed fibril texture along the new stretched direction.

The quantitative comparison of the transverse plastic deformation with experimental mechanical data showed unexpectedly good correspondence. In the real spherulitic texture of PE, the crystalline lamellae highly oriented to the draw direction, as with P in Figure 8, gave rise to the early elastic response and cavitation, while those of T in Figure 8 were highly deformable under lower stress level. The response of the real shperulitic texture may show the average of these two extreme cases, where initial small deformation is conspicuous of the largely elastic response, while the large deformation is governed by the plastic deformation of the ductile crystals.

We expect the overall average behavior of the real spherulite to be well-reconstructed from the two extreme cases of the textures. Indeed the stress–strain curve produced by the present MD simulation (Figure 11a) compares well with recent experimental measurements, such as those given by Strobl [20] with respect to the stress level of the plateau region (~15 MPa) and the degree of elongation at which the strain hardening sets in ($\varepsilon_{H}\sim 1.5)$. Lastly, the conclusions of the present paper will be summarized as follows. The melt memory was clearly shown to remain in various ordered parameters of the melt and to persist for a period of time of the Rouse time scale.

The remaining melt memory was found to give marked influences on recrystallization leading to the distinct crystalline morphologies. In addition, the distinct morphologies observed were found to cause quite different mechanical responses during their large deformation.

## Figures and Tables

**Figure 1 polymers-14-03082-f001:**
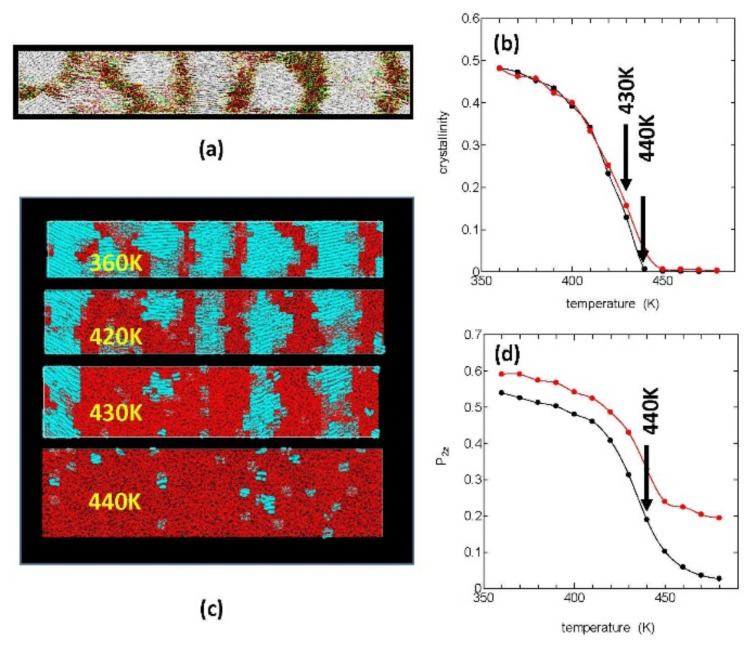
Typical melting process of the oriented stacked lamellae in terms of the crystallinity during heating at a constant rate 10 K/ns. (**a**) The initial crystalline fiber, synthesized in our previous paper. (**b**) The crystallinity is plotted vs. temperature, which shows the clear onset of melting around 440 K. (**c**) Typical melting trajectories of the orientated PE lamellae during the constant heating process at the indicated temperatures; the sample was here placed under stress free condition ($P_{xx},\;P_{yy},\;P_{zz}$) = (0, 0, 0). The blue and red regions correspond to the crystalline and amorphous regions, respectively. (**d**) The orientational order parameter $P_{2z}$ of the C-C-C chord vectors gradually decreasing but remains finite even around the melting temperature.

**Figure 2 polymers-14-03082-f002:**
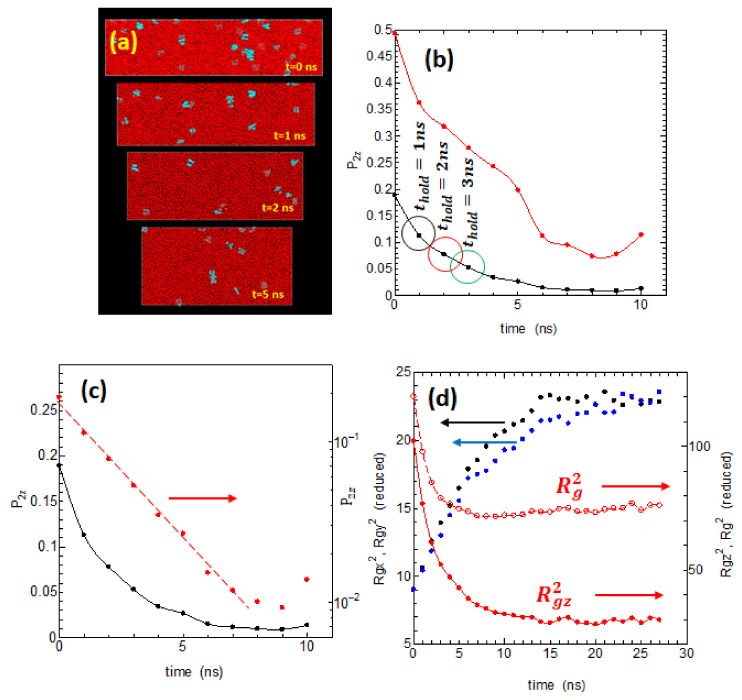
(**a**) The process of annealing the melt at 440 K (well above the melting point) vs. time of holding $t_{hold}$ at 440 K. (**b**) The loss of melt memory in terms of the order parameter $P_{2z}$ as a function of time $t_{hold}$ at 440 K. We considered the $P_{2z}$ order parameters calculated both from the chord vectors connecting C-C-C atoms (black circles) and from the end-to-end vectors of the chains (red circles). (**c**) The order parameters $P_{2z}\;$ from the chord vectors (black) and those of log ($P_{2z}$) (red) are plotted vs. time showing the clear exponential decay of $P_{2z}\;$ with time. (**d**) The changes in the radius of gyration $R_{g}^{2}$ as well as its x, y, z components, $R_{gx}^{2},\;R_{gy}^{2},R_{gz,}^{2}$ are plotted vs. time in red open circles, black filled circles, blue filled circles, and red filled circles, respectively.

**Figure 3 polymers-14-03082-f003:**
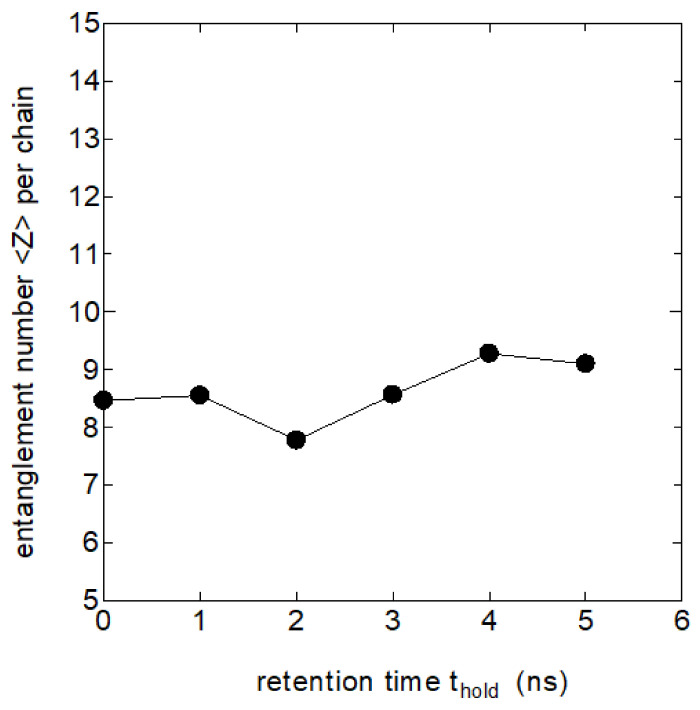
The number of entanglements per chain Z vs. the time of retention $\;t_{hold}$ at 440 K, where Z is estimated by the Z1-code by Kröger.

**Figure 4 polymers-14-03082-f004:**
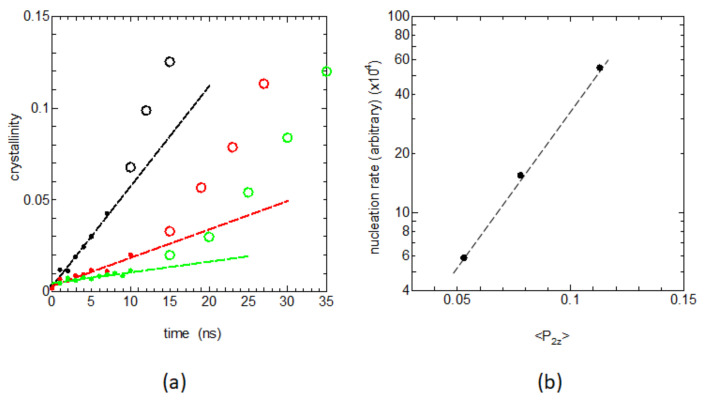
(**a**) The crystallinity increases during the initial nucleation stages for the cases $t_{hold}$ = 1 ns (black filled circles), $t_{hold}$ = 2 ns (red filled circles), and $t_{hold}\;$ = 3 ns (green filled circles), together with the data after the initial stages of nucleation (open circles). (**b**) The nucleation rates and the slopes of the initial nucleation stages are plotted as a function of the local orientational order $P_{2z}\;$ for the C-C-C chord vectors.

**Figure 5 polymers-14-03082-f005:**
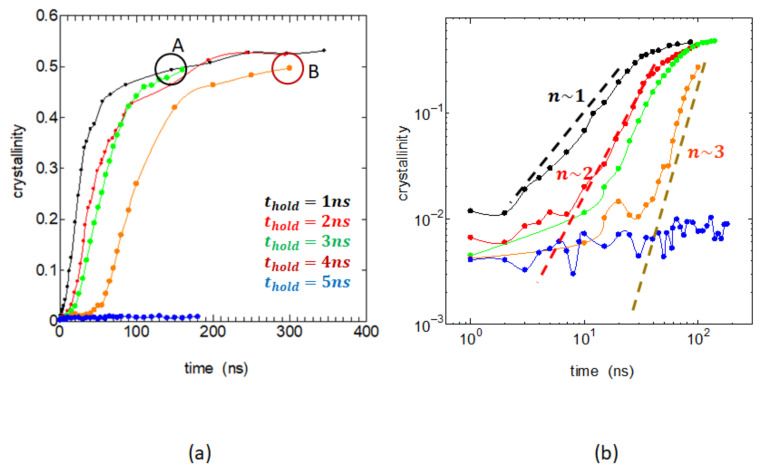
(**a**) The progress of crystallization over longer time scale of several hundred of nanoseconds for five different holding times $t_{hold}$: 1 ns (black), 2 ns (red), 3 ns (green), 4 ns (orange), and 5 ns (blue). States A and B circled represent two typical states of different crystallization histories, the behavior of deformation of which will be discussed later in Figure 7b. (**b**) The same crystallinity data are plotted in log–log scales, which are the so-called Avrami plots. Considerable changes in the slope of the linear increases are observed.

**Figure 6 polymers-14-03082-f006:**
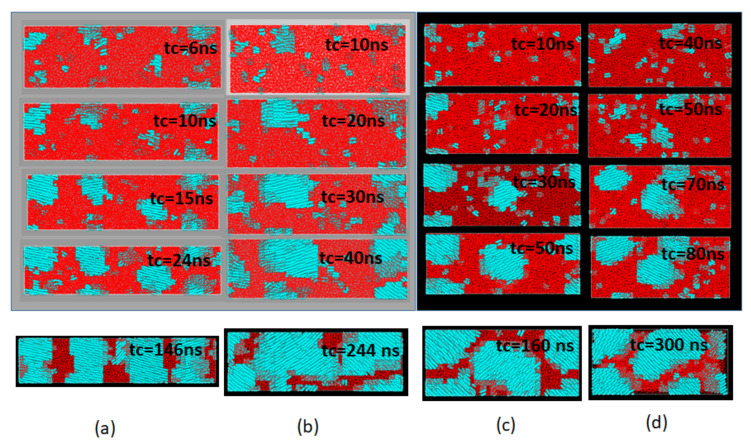
Typical morphological changes during the initial growth of the crystallites for different holing times $t_{hold}$; (**a**) $t_{hold}\;$ = 1 ns, (**b**) $t_{hold}\;$ = 2 ns, (**c**)$\;t_{hold}\;$ = 3 ns, and (**d**) $t_{hold}\;$ = 4 ns. In each snapshot are the times $t_{c}\;$ spent for the crystallization, while the last snapshot in each column indicates the final morphology obtained after a sufficient time of crystallization. The blue and red regions correspond to the crystalline and amorphous regions, respectively.

**Figure 7 polymers-14-03082-f007:**
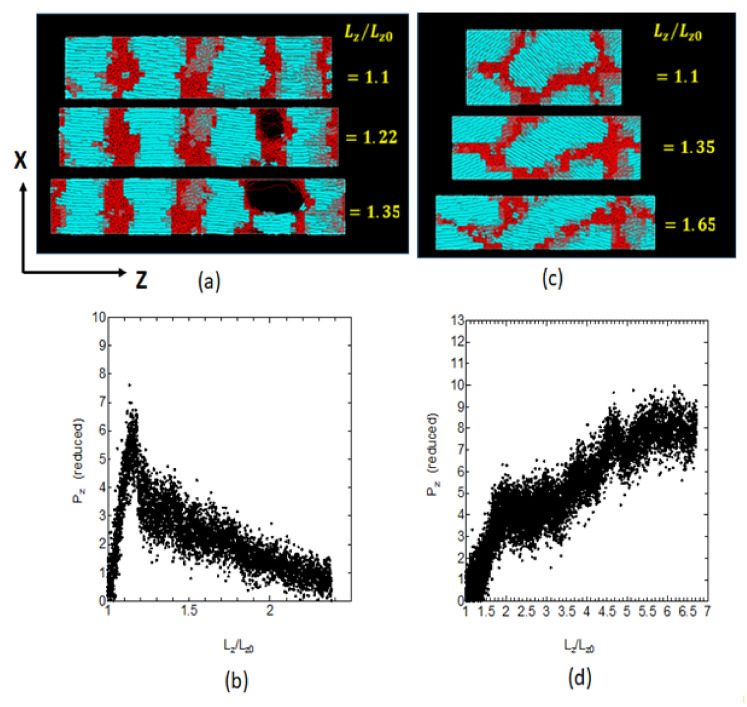
Typical fast deformation processes, with Hencky-strain rate 0.1/ns, during the uniaxial elongation along the *Z*-axis; (**a**) and (**b**) are for the well-developed textures obtained for $t_{hold}$ = 1 ns (State A in Figure 5a), and (**c**) and (**d**) are those for $t_{hold}$ = 4 ns (State B in Figure 5a). The snapshots (**a**) and (**c**) indicate the morphological changes for the indicated degree of deformation $L_{z}/L_{z0}$, while the graphs (**b**) and (**d**) are the stress–strain curves for the corresponding deformation; $L_{z}$ and $L_{z0}$ indicate the sample length along the direction Z and that of the initial state, respectively. The blue and red regions correspond to the crystalline and amorphous regions, respectively.

**Figure 8 polymers-14-03082-f008:**
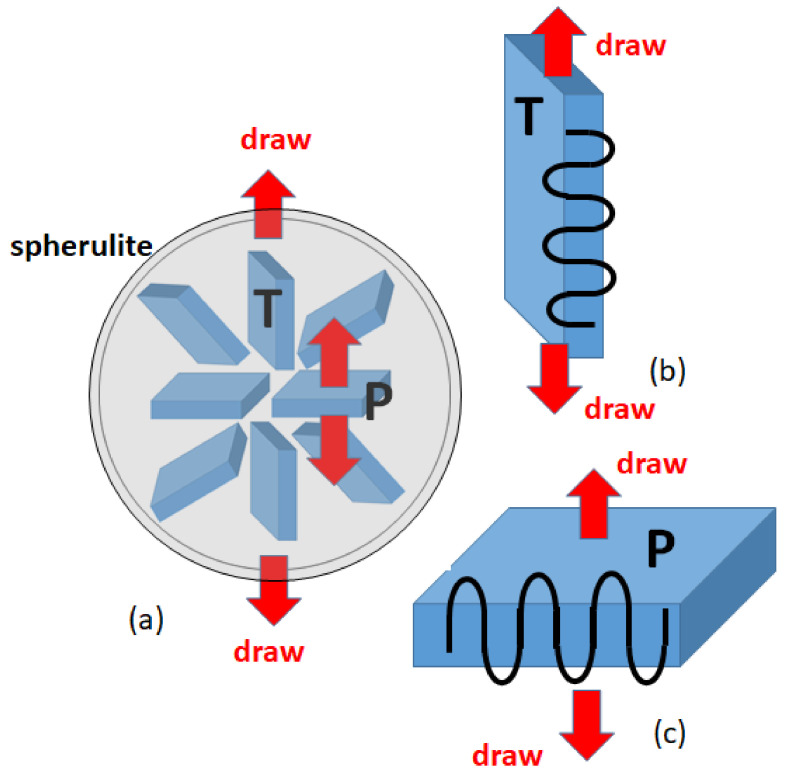
(**a**) Schematic representation of the spherulitic texture of polyethylene, where each lamella is shown as a plate extending along the radial directions; the two representative lamellae are denoted as T-type and P-type. (**b)** and (**c**) The mode of lamella deformation is along the directions transverse and parallel to the chain axes for the T-type and the P-type lamellae.

**Figure 9 polymers-14-03082-f009:**
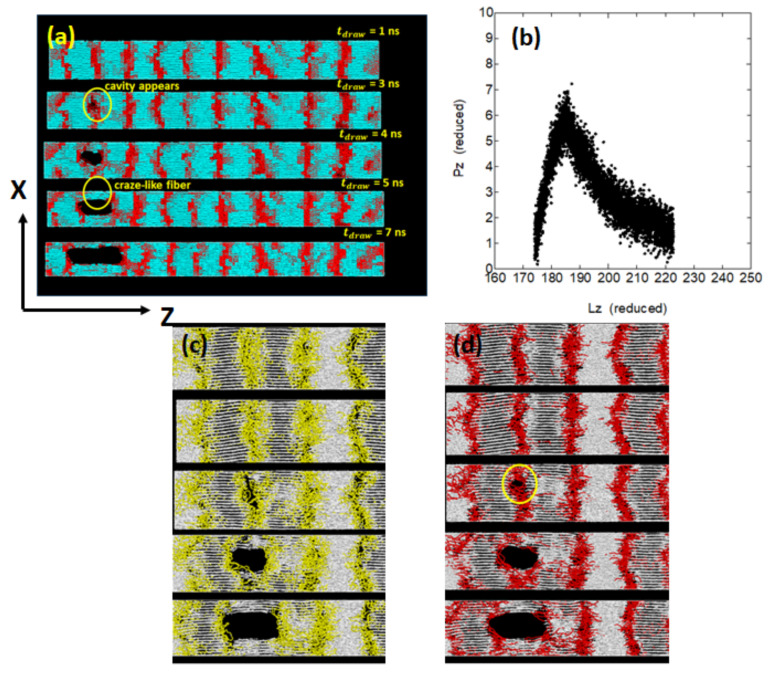
Uniaxial deformation along the *Z*-axis, with the true rate 0.05/ns, for the type-P (Figure 8c) in the well-developed stacked lamellae system of 150,000 united atoms, (**a**) snapshots during the deformation at 1~7 ns, where the blue and red regions correspond to the crystalline and amorphous regions, respectively, (**b**) the stress–strain curve, and the conformational changes in (**c**) the fold segments and (**d**) the ties segments, where white, yellow, and red segments correspond to the crystalline, fold, and tie chains, respectively.

**Figure 10 polymers-14-03082-f010:**
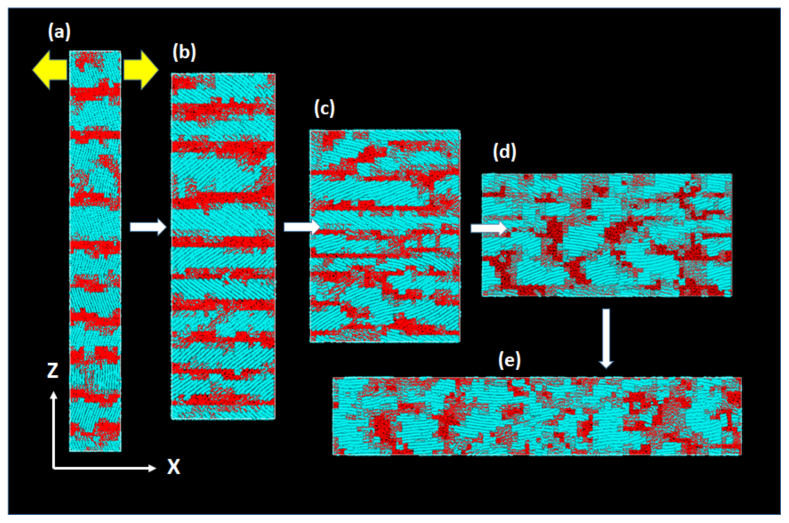
Uniaxial deformation along the *X*-axis, with the true rate 0.05/ns, for the type-T (Figure 8b) in the well-developed stacked lamellae system of 150,000 united atoms. The times (and the Hencky-strain) for the deformation are (**a**) 1 ns (0.05), (**b**) 10 ns (0.5), (**c**) 20 ns (1.0), (**d**) 30 ns (1.5), and (**e**) 40 ns (2.0). The blue and red regions correspond to the crystalline and amorphous regions, respectively.

**Figure 11 polymers-14-03082-f011:**
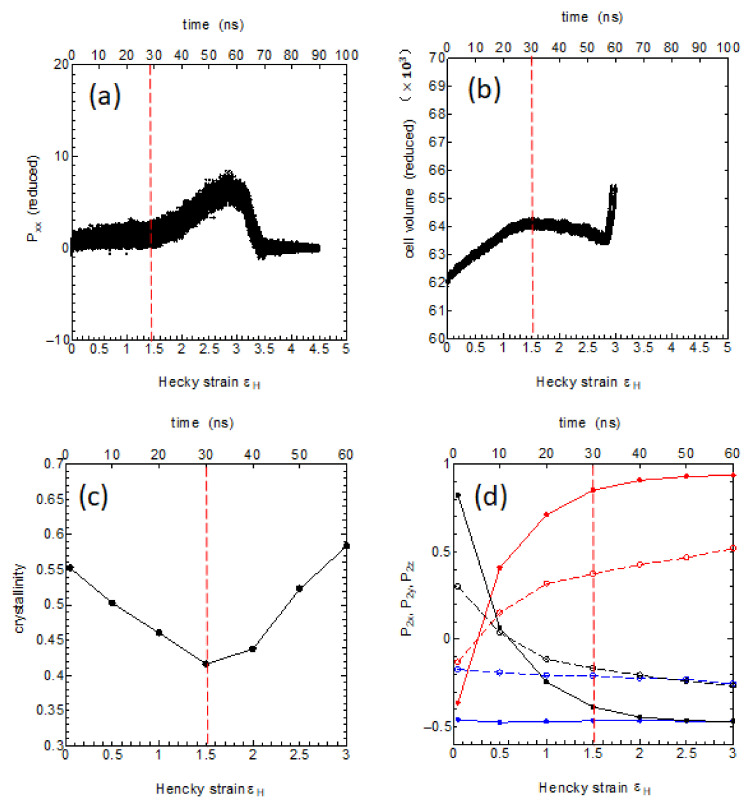
Analyses of the structural changes during the large T-type deformation of Figure 10 using various order parameters vs. $\varepsilon_{x}$, (**a**) the stress–strain relation between the stress $P_{xx}$ and the Hencky-strain $\varepsilon_{x}$, (**b**) the MD cell volume, (**c**) the crystallinity $x_{c}^{mesh}$, and (**d**) in the orientational order parameters $P_{2x},\;P_{2y},\;P_{2z}\;$ along the x-, y-, z-axes, in red, blue, and black, respectively; the filled circles with solid lines are those for the crystalline domains while the open circles with dashed lines are those for the amorphous domains.

**Table 1 polymers-14-03082-t001:** Values of the parameters used in the simulation.

Parameters	Values	Units
*m*	14	g/mol
ε (reduced energy)	500	J/mol
σ (reduced length)	0.38	nm
*r* _0_	0.4	σ
*k_b_*	10,000	ε/σ^2^
θ_0_	70.5	degree
*k* * _θ_ *	1000	ε
*k*	18	ε
*a* _0_	1.0	
*a* _1_	1.31	
*a* _2_	−1.414	
*a* _3_	−0.3297	
*a* _4_	2.828	
*a* _5_	−3.3943	

reduced pressure is 15.13 MPa.

## Data Availability

Not applicable.

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
