# Peer review of "Material Property Recovery by Controlling the Melt Memory Effects on Recrystallization and on Crystal Deformation: An Approach by the Molecular Dynamics Simulation for Polyethylene"

_polymers, 2022, doi:10.3390/polym14153082_

Round 1

Reviewer 1 Report

Title: Materials property recovery by controlling the melt memory effects on recrystallization and on crystal deformation: An approach by the molecular dynamics simulation for polyethylene

The referee would like to recommend this work to major revision and to be published after consideration according to the comments below:

1.      The writing of this paper needs to be improved and polished. Some clumsy and neglectful expressions can be found.

2.      Please add more current papers in the literature and improve introduction section. Some interesting papers related to the topic of this manuscript could be:

Experimental and numerical study on HDPE/SWCNT nanocomposite elastic properties considering the processing techniques effect. Microsystem Technologies, 26, 24232441.

Development of efficient size-dependent plate models for axial buckling of single-layered graphene nanosheets using molecular dynamics simulation. Microsystem Technologies, 24(2), 1265-1277.

3.      Some Figures (Figure 1. (a), Figure 7). and some Equations (Equation (1)-(3)) are illegible and unclear. Please correct them.

Reviewer 2 Report

Overall, this paper present a good finding that is worth for publication, however, some improvement is necessary before this paper is ready to be published.

Abstract:

-              The abstract should consists of introduction, objective of study, methodology used, main findings (quantitative information is highly recommended), and conclusion. Please make sure all this item present in the abstract.

-              Please add more specific objective for this study in the abstract. The objective of study should clearly mention the type of materials used in this study.

-              Describe the type of modelling/test conducted.

-              The main contribution/significant finding should be highlighted in the results.

-              Author might consider to reduce the usage of “we” in the abstract.

-              Please check for grammar error.

Introduction:

-              Some of the sentence used are inappropriate for scientific papers such as “What is the molecular mechanism of retaining melt memory of previous history? How does it affect the material’s recrystallization during molding process?”. Please write a passive sentence.

- Author should highlight some information regarding the material used in this study, and why this material were chosen at the first place.

-              At the end of the introduction, the author should highlight the gap of study between previous and the current study which trigger this research to be carried out, this can be followed by the general objective of this study.

-              Please check for grammar and punctuation error.

Methodology:

-              Please reorganize the position of table and equation to the center.

-              Please check for grammar error.

Results and Discussion

-              Figure caption for Fig 1 need to be simplify.

- The discussions provided were quite detail, however, it is lacking supporting argument from literature. It should be supported with appropriate reference.

-Again, author should use passive sentence in writing technical paper, usage of question mark is nor appropriate to be used.

 Conclusion

-          Conclusion section should be revised and not to be merged with discussion.

- The conclusion should be simplified and summarize into one paragraph and important finding should be highlighted.

Reviewer 3 Report

In my opinion, the article: ”Materials property recovery by controlling the melt memory effects on recrystallization and on crystal deformation: An approach by the molecular dynamics simulation for polyethylene” is well structured. The simulation activity and the used models can be considered reliable and compatible with the aim of the study. Also the outcomes seem to be quite reliable and can also be considered good, since the described phenomena have been already observed in the past. What is not so clear is whether an experimental activity has been carried out to demonstrate the convergence between results obtained from the models application and experimental data. English is quite good and the text can be easily understood, despite the text (in particular in the abstract) is arranged in a first person mode and this is not so “graceful” according to a scientific base point. Probably a text arranged in a passive (third person) form could be preferred. For this reason, the article deserves to be published after a minor revision.

Here few question:

·         Pag. 2 of 17: The authors claim: “…the entanglements in the original crystalline texture
remain even after melting and they are considered to give great influence on recrystallization. But here again experimental evidence is difficult to obtain.”

- This conclusion can be accepted  and represents a critical issue to investigate, indeed a preliminary melting is in general promoted to eliminate the thermal (and process) history of the investigated system, as well as to promote the removal of the previous ordered structures. This topic is considered as a stronghold for thermal analysis as in DSC analysis.

- The authors are recommended (this is not mandatory) to add sentences to explain the mentioned difficulties;

·         Pag. 2 of 17: The authors claim: “Our goal is to have the unified understanding of recrystallization and higher-order structure formation, and to elucidate the origin of the reduced mechanical properties in recycled materials.

- Also this topic represents a very critical issue, as the decrease in the PE mechanical properties not only depend on the applied recycling manufacturing process chain, but could significantly depend on the service life condition of the pristine components. For this reason a direct comparison between the properties of a virgin material and a recycled one could be misleading;

- The authors are recommended to clarify the basis of comparison;

·         Pag. 4 of 17 – The authors claim: “Since this argument is based on molecular level…”;

- The authors are invited to replace the term “argument” with “topic” or ”subject”;

·         Pag. 4 of 17 – Paragraph 3.1: The authors have to clarify whether the outcomes of the simulation have been experimentally verified. Despite an experimental verification could be difficult or time consuming, methodologies can be applied. In example, hot stage based devices, coupled to optical microscopes, can be used to carry out (qualitative at least) verification of the model reliability in the estimation of removal and nucleation of ordered structures for polyethylene.

·         Additionally, a description on the experimental methodology carried out to verify the obtained results is not reported or it is not so clear if it has been actually carried out. Probably many results can’t be verified with experimental test, but the authors are invited to justify their approach.
